# Effectiveness of New Tools to Define an Up-to-Date Patient Safety Risk Map: A Primary Care Study Protocol

**DOI:** 10.3390/ijerph18168612

**Published:** 2021-08-15

**Authors:** Montserrat Gens-Barberà, Cristina Rey-Reñones, Núria Hernández-Vidal, Elisa Vidal-Esteve, Yolanda Mengíbar-García, Inmaculada Hospital-Guardiola, Laura Palacios-Llamazares, Eva María Satué-Gracia, Eva Maria Oya-Girona, Ferran Bejarano-Romero, Maria Pilar Astier-Peña, Francisco Martín-Luján

**Affiliations:** 1Patient Safety and Quality Unit, Camp de Tarragona Regional Management, Institut Català de la Salut, 43005 Tarragona, Spain; mgens.tgn.ics@gencat.cat (M.G.-B.); nhernandez.tgn.ics@gencat.cat (N.H.-V.); evidal.tgn.ics@gencat.cat (E.V.-E.); ymengibar.tgn.ics@gencat.cat (Y.M.-G.); ihospitalg.tgn.ics@gencat.cat (I.H.-G.); lpalacios.tgn.ics@gencat.cat (L.P.-L.); eoya.tgn.ics@gencat.cat (E.M.O.-G.); 2Research Support Unit Tarragona-Reus, Institut Universitari d’Investigació en Atenció Primària Jordi Gol (IDIAP Jordi Gol), Institut Català de la Salut, 43202 Reus, Spain; esatue.tgn.ics@gencat.cat (E.M.S.-G.); fmartin.tgn.ics@gencat.cat (F.M.-L.); 3Facultat de Medicina i Ciències de la Salut, Universitat Rovira i Virgili, 43001 Reus, Spain; 4Primary Health Centre Doctor Sarro Roset, Primary Care Management Camp de Tarragona, Institut Català de la Salut, 43800 Valls, Spain; 5High Resolution Centre, Primary Care Management Camp de Tarragona, Institut Català de la Salut, 43840 Salou, Spain; 6Pharmacy Unit, Primary Care Management Camp de Tarragona, Institut Català de la Salut, 43005 Tarragona, Spain; fbejarano.tgn.ics@gencat.cat; 7Aragon Health Services, Primary Health Centre Universitas, 50017 Zaragoza, Spain; mpastier@gmail.com; 8Medical School, Universidad de Zaragona, GIBA-ISS-Aragón, 50009 Zaragonza, Spain

**Keywords:** patient safety, primary health care, risk management, clinical protocols

## Abstract

Background: Reducing incidents related to health care interventions to improve patient safety is a health policy priority. To strengthen a culture of safety, reporting incidents is essential. This study aims to define a patient safety risk map using the description and analysis of incidents within a primary care region with a prior patient safety improvement strategy organisationally developed and promoted. Methods: The study will be conducted in two phases: (1) a cross-sectional descriptive observational study to describe reported incidents; and (2) a quasi-experimental study to compare reported incidents. The study will take place in the Camp de Tarragona Primary Care Management (Catalan Institute of Health). In Phase 1, all reactive notifications collected within one year (2018) will be analysed; during Phase 2, all proactive notifications of the second and third weeks of June 2019 will be analysed. Adverse events will also be assessed. Phases 1 and 2 will use a digital platform and the proactive tool proSP to notify and analyse incidents related to patient safety. Expected Results: To obtain an up-to-date, primary care patient safety risk map to prioritise strategies that result in safer practices.

## 1. Introduction

### 1.1. Patient Safety, the Cornerstone of Quality of Care 

The primum non nocere principle included in the *Hippocratic Oath* and attributed to Galen constitutes one of the basic principles of medicine. It is understood that any medical intervention might cause harm, and a reasonable risk–benefit balance is accepted in medical interventions [1]. However, a tendency to overly justify and accept adverse effects as unavoidable has persisted among health care professionals. Fortunately, the relevance of patient safety (PS, the limitation of harm related to health care interventions) has steadily gained momentum [2]. Indeed, PS became a priority among health policies after the 1999 publication of the report “To Err is Human” by the Institute of Medicine [3], which revealed that in the USA, medical errors constituted the eighth leading cause of mortality. In Spain, the Ministry of Health, Social Services and Equality, in collaboration with the Autonomous Communities, developed a PS strategy for the National Health System. Line three of the reviewed strategy introduced risk management and systems to notify and learn from incidents [4]. Following publication of the strategy in 2015, most autonomous communities have developed systems for the communication of PS incidents [5].

In 2012, a pilot of the project Functional Units for PS was conducted in our health region within the framework PS Strategic Plan in Primary Care, launched in 2011 by the PS Promotion Service, which emerged in Catalonia in 2009 with the collaboration of the Ministry of Health and the Catalan Department of Health [6,7]. The plan aimed to design, implement and evaluate strategies to improve PS in Primary Care using reactive and proactive approaches. The main reactive strategy was the activation of an incident notification system in May 2013. The World Health Organization (WHO) considers PS incidents as events or circumstances that caused or could have caused unnecessary harm to a patient while receiving health care [8]. PS incidents can be divided into (a) incidents that do not reach the patient (reportable circumstances, near-misses); (b) incidents that have not caused any harm; and (c) incidents that have caused harm, i.e., adverse events.

### 1.2. Systems to Notify and Learn from Incidents 

An effective reporting system is the foundation for safe practices and for establishing a culture of safety [9]. In fact, all organisations are responsible for reducing PS incidents, and notification should be the aim as a preliminary step to reduce the incidence [10]. The analysis of PS reports, and of adverse events specifically, should provide insight on the risks associated with health care and should suggest preventative strategies [11]. As such, we believe that voluntary incident notification systems are a crucial source of information, and that, although standardised and validated methods to rate incidents have not yet been established, a combination of proactive and reactive approaches should be encouraged [12]. However, these systems are notoriously underused, because, while notification depends on awareness and willingness to report incidents, it is the responsibility of organisations and leaders to avoid finger pointing and encourage the reporting of incidents as an opportunity to increase PS [13]. To encourage professionals to report, they must also perceive how their notifications result in the implementation of improvement strategies [14]. In the United States and Europe, notification systems have been operational from the second half of the 1990s. However, these systems vary regarding their mandatory/voluntary nature, types of reported incidents, internal/external management of incidents, and analysis and dissemination of the data generated [15].

### 1.3. Patient Safety in Spanish Primary Health Care Services

Interestingly, although primary care is the cornerstone of health services and where most patients are visited and treated, the literature on PS is mainly hospital-based [16]. A systematic review published by the WHO to determine the frequency, severity and preventability of PS incidents underscored the lack of large scale robust studies in primary care [15]. A recent review has emphasised the gap between PS studies conducted in hospitals versus primary care services (in this review only 4% (*n* = 3) of studies had been conducted in primary health care services) [17]. 

In Spain, the benchmark for PS in primary care is the 2008 APEAS study (Spanish acronym of Patient Safety in Primary Healthcare), which analysed over 96 thousand encounters, and concluded that 1 in 100 visits was associated with a preventable adverse event [18]. The APEAS results showed that events related to medication (adverse events and errors in medication) were the most commonly notified, although there were also notifications related to communication, management and organisational issues [19]. It is a cause for concern that despite an increasing emphasis on the prevention of PS incidents and harm minimisation protocols, relevant aspects of PS epidemiology remain unexplained. Moreover, we believe that the pervasive use of new technology and changes in the type and number of visits have driven a change in the epidemiology of notified PS incidents. A recently published systematic review recommends a periodic evaluation of PS incidents in primary care [20].

### 1.4. Aim of the Project

The objective of this project is to describe and analyse the incidents collected proactively and reactively to design the risk map in a region with a prior strategy to improve PS.

## 2. Materials and Methods

### 2.1. Study Design

The study will consist of two phases of analyses:

Phase (1)—Cross-sectional descriptive observational study with analysis of reported incidents for one year, from January to December 2018. 

Phase (2)—Quasi-experimental study comparing the reported incidents with incidents reported in the APEAS study [18], using notifications from the 2nd and 3rd weeks of June 2019 (similarly to the APEAS study).

The results obtained during Phases 1 and 2 will prompt discussion with a focal group of experts, with the ultimate aim to draw the risk map.

This protocol describes Phases 1 and 2.

### 2.2. Study Setting

All Catalan Institute of Health primary care centres (PCCs) of the Camp de Tarragona region, encompassing a population of 328,945 patients, 20 Primary Care Teams, 2 Sexual and Reproductive Health Care Teams, and 2 Primary Care Emergency Centres, will be studied.

### 2.3. Recruitment Process

During Phase 1, all notifications collected for one year (January–December 2018) will be analysed. In Phase 2, similarly to the APEAS study, the notifications reported during the two central weeks of June 2019 will be analysed; adverse events will be separately examined.

All health centres enter the information on PS incidents using a structured form accessible from the TPSC (The Patient Safety Company)™ Cloud platform [21] on the corporate intranet of the Tarragona Regional Management of the Catalan Institute of Health (http://camptarragona.cpd2.grupics.intranet/). This information is automatically stored (including a backup copy). Access to the management system is password-protected.

### 2.4. Eligibility Criteria 

All incidents reported by professionals of PCCs belonging to the Catalan Institute of Health Camp de Tarragona, through a digital platform (allows notifications, analyses and improvement actions), plus incidents identified with proactive activities defined in other PS strategies.

Eligibility criteria are as follows: Inclusion: all incidents reported during the study period. Exclusion: failure to describe true PS incidents; duplicate reports; notifications of known adverse drug reactions; incidents with insufficient information for correct coding; and evaluation adjudication pending analysis.

For Phase 1, participation will include all PCCs of the Catalan Institute of Health Camp de Tarragona, with 1400 professionals, with a similar proportion of physicians, nurses, nursing assistants and admission/administrative staff. Phase 2 will require the voluntary participation of all professionals (doctors, nurses and administrative staff) from the 20 centres included. On average, each physician visits 30 patients per day, which amounts to approximately 45,000 patient visits in 2 weeks (10 working days). Considering that adverse events are reported in 3% of primary care consultations [20], approximately 1350 PS incidents are described, a number considered sufficient to establish a risk map.

### 2.5. Data Collection Tools

#### 2.5.1. Digital Platform for the Notification and Analysis of PS Incidents

The notification of incidents is voluntary, confidential and anonymous in the digital platform implemented as a reactive strategy by the PS Functional Units Project. First, notifications are received and managed by the Quality and Safety officer of the PCC of the notifier, and later in the Regional Quality Unit by two specifically trained professionals responsible for the collection of all regional PS incidents. This tool has been available since 2013. All professionals are encouraged to notify any PS incident detected on the TPSC_Cloud platform at any time. For Phase 1, we will collect the data introduced in this platform from January to December 2018.

#### 2.5.2. Incidents Detected Using the Proactive Tool proSP

Incidents will also be reported using the proSP (proactive in Patient Safety) tool that automatically manages checklists of different processes (daily review of the emergency equipment, temperature control of refrigerators, management of supplies, pharmacy stores, etc.). All errors detected in the periodic review send an automated email warning to the Quality and Safety officer, the process manager and the primary care centre director. Additionally, information originated from PS indicators and type of incidents is updated daily. 

A proactive search for incidents will occur during 10 days in Phase 2. After each visit, professionals will assess whether the patient has suffered an adverse event; if so, data related to the incident will be recorded on the platform.

Both tools can be accessed from every computer in participating PCCs.

### 2.6. Study Variables

The possible values of the variables analysed are:Type of incident according to WHO [2]: (a) falls and other accidents; (b) patient behaviour; (c) health care teams; (d) analogue and digital documentation; (e) clinical management and procedures; (f) clinical-administrative management; (g) health-care-associated infection/severe pressure ulcers of nosocomial origin; (h) infrastructures, premises or facilities; (i) medication and (j) blood products;Type of incident according to Catalan Health Department Accreditation Model [22]: (a) urgent care; (b) continuity of care; (c) health education; (d) ethics and rights of citizens; (e) management of clinical supplies; (f) waste management; (g) laboratory; (h) administrative processes; (i) care process; (j) imaging services; (k) general services (cleaning, etc.); (l) social work; (m) safe use of medication; (n) vaccines; (o) surveillance; (p) prevention and (q) infection control;Causal factors according to APEAS study [18] (see Table 1).Centre: Primary Care Team; Primary Care Emergencies; Sexual and Reproductive Health Centre;Risk: very low; low; moderate; high; extreme;Severity of incident according WHO criteria [2]: reportable circumstance (opportunity to cause harm, with no incident); near-miss incident (incident does not reach the patient); no harm incident (incident reaches the patient without evidence of harm); incident with harm—adverse event (incident causes harm to a patient);Severity of damage according International Classification for PS in Primary Care [10]: (a) circumstance with opportunity for error; (b) error occurred but detected before reaching the patient; (c) error not resulting in injury; (d) patient requires observation but no injury has occurred; (e) treatment required or temporary injury produced; (f) hospitalisation lengthened and temporary injury caused; (g) permanent injury caused; (h) near-death situation produced; (i) has caused or contributed to the death of the patient;Contributing factors: environment; external; organisation; patient; professional;Quality of notification: correct or unclear;Who reviews the notification: primary care team; PS Functional Unit; responsible within Primary Care Management; responsible within the Department of Health; other;Safe practices of the PS Functional Unit: improvement team; bulletin; warning; committee; document review; training;Primary Care Team safe practices: improvement teams; training; committee/management, review of documents;Status: solved; pending decision by regional PS Functional Unit; pending decision by Primary Care Management; pending decision by other Units.

### 2.7. Response Variables 

Incident, Adverse Event and Avoidable Adverse Event.
Incident: unanticipated random event related to the health care process that does not cause harm to the patient;Adverse event: unanticipated accident identified during the consultation, which has caused injury and/or disability, derived from the health care process and not from the patient’s underlying disease;Avoidable adverse event: the PS Unit, constituted by the management of the Regional Quality Unit together with the responsible professionals within each Primary Care Centre, have to agree on avoidable adverse effects. Incidents are scored on a scale of 1 to 6 (1 = no evidence and 6 = evidence of avoidability). Events with a score equal or greater than 4 are considered avoidable;Severity of the event: an adverse event is considered severe if it causes death, disability or requires surgical intervention; moderate if it results in a hospital stay of at least one day or if it requires emergency or specialised care; and mild when the incident does not cause any of the above.

The Regional Quality Unit, together with the responsible professionals within each centre and the research team, carries out a peer-review of the reported incidents, with the objectives of: (a) validating their inclusion in the study; (b) evaluate avoidability; (c) classify type of event; and (d) classify causal factors.

### 2.8. Data Analysis

A descriptive analysis of categorical and quantitative variables will be conducted. For the bivariate analysis, we will use the chi-squared test or Fisher’s test for categorical variables and Student’s *t*-test or ANOVA for quantitative variables. We will subsequently perform logistic regression for the multivariate analysis, separately analysing the dependent variables incident, adverse event and avoidable adverse event. The models will be adjusted for the main confounding variables.

For comparison with the APEAS study [18], we will only consider adverse events and exclude unharmful incidents. We will perform a comparison of proportions and chi-squared test for the variable causal factors according to APEAS and for this variable in relation to the professional category of the notifier. The level of significance will be set at *p* < 0.05. The statistical package R (R Foundation for Statistical Computing, Vienna, Austria. 2018), version 3.4.4, will be used for all analyses.

### 2.9. Expected Results

Evidence-based PS strategies based on analysis of incidents and the construction of a risk map are urgently needed. This project aims to promote PS in Primary Care, encouraging notification of incidents and a culture of patient safety among professionals. The analysis of notifications and of adverse events will define an up-to-date risk map, which will drive improvement strategies for safe practice. These results may extrapolate to other regions and contribute to multicentre studies to further this research.

## 3. Discussion

Two prior studies are considered pivotal in Spain regarding the epidemiology of adverse events: the ENEAS 2005 study (National Study on Adverse Events linked to Hospitalization) [23], and the 2008 APEAS study [18] on adverse events in primary care. Later publications focus almost exclusively on adverse events in the hospital setting.

Several groups from the USA, Canada, Australia, Holland, the United Kingdom, Kuwait, and China, investigate patient safety, and some specifically in primary care [24,25,26,27,28,29,30,31,32]. Using a methodology similar to the APEAS study in France, Michel P et al. [33] observed a higher incidence of adverse effects (26 per thousand consultations per week), mostly attributed to organisational and communication problems. A systematic review on PS in primary care concluded that the majority of harmful incidents relate to diagnosis and prescription [15]. The National Institute for Health Research (NIHR) published an extensive analysis of incidents reported from primary care [34] and concluded that most relate to diagnosis, emphasising communication among the main contributing factors. The same authors have published a protocol to study the epidemiology of adverse events in primary care [35].

In Spain, the most important study on the impact of adverse events in primary care is the APEAS study (study on the safety of patients in primary health care) [18], a prevalence, observational study conducted in 48 primary care centres from 16 autonomous communities during two weeks of June 2007. This study, which included 96,047 patients, showed a prevalence of adverse events of 11.18 ‰ (95% CI %: 10.52–11.85). The study stressed the importance of preventing adverse events in primary care because, despite a seemingly low incidence, the absolute number of affected patients is high (a physician with 20 consultations per day could have a harmful incident per week). Of note, the study concluded not only that up to 70% of adverse events could be avoided, but that severity and avoidability were positively correlated, suggesting that 80% of severe incidents were considered preventable with appropriate measures. 

After the APEAS study, no significant publications on PS in primary care in Spain have been published. The objective of the current study is to determine the effectiveness of the implementation of various PS reactive and proactive tools to define an up-to-date primary care PS risk map based on a large sample of incidents. To this end, we will firstly describe the incidents reported using the various PS strategies that should identify critical areas, and secondly, we will analyse the reported incidents and compare them with the incidents in the APEAS study [18]. Finally, an external committee of experts will validate the risk map and the suggested PS improvement strategies.

### 3.1. Limitations

This study contains methodological limitations. Principally, the results will originate from a voluntary reporting system, directly introducing an underreporting bias, since a considerable number of incidents may not be reported. Additionally, we will not consider other relevant data sources such as complaints, which could affect the internal validity of the study. However, active notification of PS incidents is considered a more common and affordable method [35]. We believe that the calculated sample will produce a representative PS risk map.

The quality of records should also be considered [36]. We expect to obtain reliable outcomes, since the study will be conducted in a health region where PS improvement strategies have been prioritised and incidents are peer-reviewed and validated by PS Functional Unit experts. We anticipate obtaining outcomes that can be extrapolated and lead to future collaborations. 

### 3.2. Ethics and Dissemination 

This study protocol was approved by the Clinical Research Ethics Committee (CEIC) of the Primary Care Research Institute (IDIAP) Jordi Gol (July 2019, 199/126-P). The study will follow the tenets of the Declaration of Helsinki and Good Clinical Practice, as stated in the IDIAP Jordi Gol Guide to Good Research Practices in Primary Care. All participating researchers and associates will sign a collaboration agreement in which they undertake to abide by good clinical practice standards.

## 4. Conclusions

This study will provide a comprehensive, real snapshot of patient safety concerns in primary care and suggested improvement strategies in Catalonia.

## Authors Contributions

Conceptualization, M.G.-B., N.H.-V., and E.V.-E.; methodology, M.G.-B. and F.M.-L.; validation, M.P.A-P., F.M.-L. and C.R.-R.; formal analysis, M.G.-B. and F.M.-L.; investigation, N.H.-V., E.V.-E., Y.M.-G., I.H.-G., F.B.-R., E.M.O.-G. and L.P.-L.; resources, L.P.-L., Y.M.-G., and F.B.-R.; data curation, E.V.-E., Y.M.-G., I.H.-G., E.M.S.-G., E.M.O.-G. and F.B.-R.; writing—original draft preparation, M.G.-B., N.H.-V., E.M.S.-G., M.P.A.-P., F.M.-L. and C.R.-R.; writing—review and editing, M.G.-B., F.M.-L. and C.R.-R.; visualization, N.H.-V., L.P.-L., F.M.-L. and C.R.-R.; supervision, M.G.-B., F.M.-L. and C.R.-R.; project administration, Y.M.-G. and I.H.-G.; funding acquisition, M.G.-B., N.H.-V., I.H.-G. and F.M.-L. All authors have read and agreed to the published version of the manuscript.

## Figures and Tables

**Table 1 ijerph-18-08612-t001:** Causal factors according to APEAS study *.

Medication	Clinical Management	Communication	Diagnosis	Management
Incorrect dosage	Suboptimal management of the patient	Doctor–Patient	Diagnostic delay	Long waiting list
Lack of adherence	Inadequate technique	Doctor–Nurse	Diagnostic error	Wrong appointment
Missing dose	Inadequate management of warning signs	Doctor–Doctor Doctor–Admin officer	Delayed referral to specialised care	EHRs issues
Wrong medication		Nurse–Patient Nurse–Nurse Nurse–Admin officer Admin officer–Patient Admin officer–Admin officer		Mistake in health information
Drug interaction		Cultural barrier		Error in patient identification
Incorrect treatment duration		Language barrier		
Wrong frequency of administration		Another communication factor		
Wrong patient				
Insufficient monitorisation				
Manipulation or preparation error				
Ineffective prescription				
Other causes				

* Causal factors included in the APEAS study questionnaire [18].

## Data Availability

Data are available on resasonable request. The full dataset and statistical code are available from the first author (principal investigator) mgens.tgn.ics@gencat.cat.

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
