# Peer review of "Effectiveness of New Tools to Define an Up-to-Date Patient Safety Risk Map: A Primary Care Study Protocol"

_ijerph, 2021, doi:10.3390/ijerph18168612_

Round 1

Reviewer 1 Report

This is a very interesting project and will have a great application when it is implemented. Unfortunately, it has no data or results to justify its use, as the information has not yet been managed. In fact, the work presented does not have a "results" section. The work is written in the future tense as it has not yet been fully carried out. It is necessary to introduce a "results" section (when obtained) to know the scope of these new tools and to know if the protocol really works. There is no doubt that the idea is a good one and that, in addition, the work will analyze a large amount of data, but at present, these data have not been treated, nor has the relationship between them. We do not know variables such as its internal consistency index (if it is going to be calculated), whether it achieves contrastable and empirical objectives,... More than a manuscript, this is the presentation of a project or protocol for carrying out further research. This protocol is well structured and shows original ideas; however, it does not provide relevant or novel information because the data have not yet been analyzed. It is proposed to the authors that, if they already have information that they can contribute, this should be done at this stage of the research, even if only in a preliminary way, in order to be able to move forward. Thank you for your attention. 

Author Response

Thank you for your comments. As you point out, this work does not provide results as such, since it is an article for the protocol section of the journal.

We believe that the term questionnaire in reference to APEAS might be misguiding. APEAS is a structured form containing information on different aspects of patient safety events.  It was designed ad hoc for the APEAS study based on a formative model, and it has not been validated. In this kind of forms, according to COSMIN (COnsensus-based Standards for the selection of health status Measurement INstruments) methodology [ref], the set of items do not need to be correlated and together they form the construct.  Internal consistency in this case is not relevant. Following your comment and also COSMIN recommendations, we will explain this aspect in the revised protocol.

REF: Mokkink LB, Prinsen CAC, Patrick DL, Alonso J, Bouter LM, de Vet HCV, et al. COSMIN Study Design checklist for Patient-reported outcome measurement instruments. Versión july 2019. Available at: https://www.cosmin.nl/wp-content/uploads/COSMIN-study-designing-checklist_final.pdf 

Reviewer 2 Report

You have chosen a very interesting research topic of great importance for quality healthcare.

The justification for your research is very well done and the work you set out will be very interesting to read when you do it.

I have just a few small points to make:

- Introduction: I would like you to clarify the aim of the study. When are the data going to be taken and what are they going to be compared with?

- It would be useful to talk about the APEAS study briefly in the introduction.

- Materials and methods: I don't understand why you say phase 2 will be conducted in 2019. If you are writing a protocol you will have to put dates after July 2021.

- Table 1: there seems to be some missing data in the diagnosis and clinical management columns. If there is no data missing, it must be a misunderstanding on my part, so please clarify the table for a better understanding.

Thank you very much for allowing me to read your project. I hope that the results will improve the quality of care.

Author Response

We appreciate your comments. We will respond to each of your points, with changes in red in the revised manuscript (see attached manuscript).

In the introduction section, we have clarified the object of the study, the period in which the data will be collected and the data against which this information will be compared.

Following your suggestion, a new paragraph has been added in the introduction to briefly explain the APEAS study.

In the material and methods section,  we have explained the periods of data collection and the data against which these information will be compared. This work constitutes the protocol for a retrospective study that uses the records collected during 2018 for Phase 1, and during the 2nd and 3rd week of June 2019 for Phase 2. Additionally, the 2019 data will be compared against data from the 2008 APEAS study.

The changes requested for Table 1 have been introduced.

Reviewer 3 Report

Effectiveness of new tools to define an up-to-date patient safety risk map: a primary care study protocol
1. Introduction
More related literature and detailed explanation should be added in this part for lacking protocol of patient safety.
2. Materials and Methods
(1) Please add the full name of APEAS, ICS, TPSC, PCC at Line85, Line96, Line 99 and Line 111, respectively.
(2) 2.4. Eligibility Criteria: How to calculate that your sample size? (3) In addition to doctors (35%) and nurses (35%), who are the other 30% professionals?
(4) Please describe in more detail how to select the150 physicians in Phase 2?
And, will the patients be recruited from inpatients? or outpatients?
3. I suggest that the result should include tables or figures, it is not easy to understand what your results are. And, there is a lack of statistical analyses in this part.
4. Because the results are not very clear, I am not sure whether the connection between your results and discussion is compatible.

Author Response

We appreciate your comments. The changes introduced following your suggestions appear in green in the revised manuscript (see attached manuscript).

Point 1: More references have been added to the Introduction, as suggested (in  the document, see attached manuscript in the reference section).

Point 2 (1): The full name has been added as suggested (see attached manuscript).

Point 2 (2) : The sample size has not been calculated, all notified incidents are collected.

Point 2 (3) : Other professionals refers to administrative staff. Each primary care center has a multidisciplinary team that consists of family doctors, pediatricians, nurses, dentists, physiotherapists, psychologists, technical assistants and administrative staff.

Point 2 (4):  We have added changes in the manuscript to clarify this information.

Point 3 and 4: This article contains no results, since it is at this point a study protocol.

Reviewer 4 Report

This project is to update the primary care patient safety risk map using the description and analysis of incidents reported in a region with a prior strategy to improve patient safety; they also expect to obtain an up-to-date, primary care patient safety risk map to prioritize strategies that result in safer practices. I do have some comments as listed below in the order noted.

Comment 1:

The quality of the data set is very important, especially for the incidents reported by professionals of primary care centers. For this reason, please clarify and list the inclusion criteria and exclusion criteria of Eligibility Criteria in the Materials and Methods section.

Comment 2:

In 2.5 Data collection tools, I deem it necessary to mention when and how is each data collection tool performed.

Author Response

We appreciate your comments. The changes introduced following your suggestions appear in purple  in the revised manuscript (see attached manuscript).

Point 1:

All incidents will be included in the TPSC_Cloud system during the periods indicated (2018 (whole year), and 2019 (2 weeks in June)). See sections Aim of the project and Study Design (in red).

As suggested, we have added inclusion and exclusion eligilibility criteria in the Materials and Methods section. The changes appear in green in the revised manuscript (see attached manuscript).

Point 2:

All professionals in our health centres enter the information on PS incidents using a structured form accessible from the TPSC Cloud ™ platform on the corporate intranet of the Tarragona Regional Management of the Catalan Institute of Health (http: //camptarragona.cpd2. grupics.intranet/). This information is automatically stored (including a backup copy). Access to the management system is password protected.

This information is contained in the 2.3. Recruitment Process section of the manuscript (see in the manuscript in purple colour).

All professionals are encouraged to notify any detected PS incident on the TPSC_Cloud platform. The TPSC_Cloud tool is used to reactively notify incidents (2018) and the PRO_SP tool is used  to proactively notify incidents (fourteen days of June 2019, similarly to the APEAS study). This information is contained in the 2.5. Data collection tools.

Round 2

Reviewer 1 Report

Thank you very much for your work and efforts to improve the manuscript. Regarding suggestions, it is still advisable to expand the theoretical framework (the introductory section), so that the theories, currents, authors and previous publications that can justify the implementation of the protocol and the variables involved are clear. Thank you. 

Author Response

We appreciate your comments and suggestions. For our team, your opinions have a lot of value and we are sure that your contributions will help improve our manuscript.

As you have suggested in your review, in the current document we have expanded the theoretical framework in the introduction section, you can see the changes in blue, as well as we have expanded the bibliography (too in blue).

We appreciate your opportunity and we are at your disposal for any other suggestions. 

Reviewer 3 Report

This article provided new tools to define an up-to-date patient safety risk map to prioritize strategies that would result in safer practices. 

More importantly, this study concluded that using "new tools to define an up-to-date patient safety risk map" can avoid adverse events as well as make severe incidents to be preventable with appropriate measures.

Author Response

Thank you very much for your evaluation and comments. Our research team appreciates your contributions. It help to improve our work.